# The LIV-1 Subfamily of Zinc Transporters: From Origins to Present Day Discoveries

**DOI:** 10.3390/ijms24021255

**Published:** 2023-01-09

**Authors:** Kathryn M. Taylor

**Affiliations:** School of Pharmacy and Pharmaceutical Sciences, Cardiff University, Redwood Building, King Edward VIIth Avenue, Cardiff CF10 3NB, UK; taylorkm@cardiff.ac.uk

**Keywords:** SLC39A, zinc transport, LIV-1 family, ZIP7, ZIP6, ZIP10, LIV-1, LIV1

## Abstract

This review explains the origin of the LIV-1 family of zinc transporters, paying attention to how this family of nine human proteins was originally discovered. Structural and functional differences between these nine human LIV-1 family members and the five other ZIP transporters are examined. These differences are both related to aspects of the protein sequence, the conservation of important motifs and to the effect this may have on their overall function. The LIV-1 family are dependent on various post-translational modifications, such as phosphorylation and cleavage, which play an important role in their ability to transport zinc. These modifications and their implications are discussed in detail. Some of these proteins have been implicated in cancer which is examined. Furthermore, some additional areas of potential fruitful discovery are discussed and suggested as worthy of examination in the future.

## 1. Introduction

This review describes features of the LIV-1 family of zinc transporters from a historical perspective. It mentions how the LIV-1 family was originally discovered from the conserved zinc-binding motif seen in the LIV-1 sequence, now known as SLC39A6 or ZIP6. It then describes the main differences between the LIV-1 family of proteins and the other ZIP family transporters. This review then considers aspects of post-translational modification and how these have a role in the location and activity of these transporters. Initial investigations of this family of proteins often used recombinant proteins and the failures and successes of this technique are discussed, pointing out some peculiarities in ZIP6 and ZIP10 expression. As more is discovered about the function of these zinc transporters, a number of the LIV-1 family members are now implicated in cancer and details of their roles in this disease is mentioned. This review finishes by mentioning areas where further work and new discoveries may be especially fruitful in the future.

## 2. The Origin of the LIV-1 Family of Zinc Transporters

The LIV-1 gene has been recognized as an estrogen-regulated gene since 1994 [1] and implicated in metastatic breast cancer due to its increased presence in lymph nodes [2]. LIV-1 was one of four genes discovered at Liverpool University with potential roles in breast cancer and termed LIV-1–4. The detection of LIV-1 was associated with estrogen receptor-positive breast cancer [3] and with the metastatic spread of these cancers to the regional lymph nodes [2]. However, there was no indication of the function of the LIV-1 protein in cells until the protein sequence was examined some years later. LIV-1 is now called ZIP6 or SLC39A6 as a member of the SLC39A family of zinc transporters and the subfamily that it originates from is called the LIV-1 subfamily and contains nine sequences (Figure 1).

### Discovery of the LIV-1 Subfamily

The protein sequence of ZIP6 was shown to contain many histidine residues, known to bind zinc, as well as be predicted to contain eight transmembrane domains (TM) [4]. Furthermore, a unique motif, HEXPHEXGD, consistent with the zinc-binding active site motif of zincin and PDF metalloproteases, HEXXHXXG, [5,6,7] was discovered in TM5. This information was used to search the non-redundant NCBI database using BLAST and a number of other potential zinc transporters belonging to the same subfamily were identified [8] and termed the LZT family. This nomenclature was short-lived as in the same year this family of zinc transporters were officially named the SLC39A family [9], a solute carrier family, which they are known as today. This conserved HEXPHEXGD motif was designated as the signature motif for the LIV-1 subfamily of ZIP transporters, dividing the 14 human members of the ZIP transporters into four families (Figure 1). There are nine human family members in the LIV-1 subfamily and it is mainly these transporters that are implicated in diseases, suggesting an important role for these zinc transporters which has not yet been fully elucidated.

## 3. Special Features of the LIV-1 Subfamily of Zinc Transporters

Analysis of the protein sequences of these zinc transporters has allowed much information to be obtained about the LIV-1 family which may be relevant to their function in cells. The nine human members of this LIV-1 subfamily of ZIP transporters are ZIP4-8, ZIP10 and ZIP12-14 [10]. They are all predicted to have eight transmembrane domains [10], which has been confirmed by structural data [11], although many transmembrane prediction packages do not recognize TM4 and TM5 as valid transmembrane domains due to their low hydrophobic quality, many of which still list the total number of transmembrane domains as six or less. Additionally, these TM domains contain histidine and proline residues which were not considered valid for TM domains in the early algorithms used for TM prediction. This means that internet protein pages such as Uniprot list an incorrect number of TM domains, usually only five or six TM domains, which should be treated with caution when interpreting results. These nine sequences have been aligned using the Clustal Omega software [12] (Figure 2) and shaded using Boxshade where the black residues are identical and the grey residues are from similar amino acid families. Generally the transmembrane domains are the most homologous areas of the sequences, consistent with these proteins having a role in transporting zinc. However, there are additional features present in the LIV-1 family that are absent in the rest of the ZIP family of zinc transporters (Figure 3) which are discussed in more detail below.

### 3.1. HEXPHE Motif

The HEXPHE motif in TM5 is highly homologous, indicative of its importance to the protein function of this family. This motif fits the catalytic zinc-binding site motif of the zincin and PDF groups of metalloproteases, HEXXHXXG, which is essential for their proteolytic activity [5,6,7]. Both these families of metalloproteases have been well characterized for many years and are known to be involved in many diverse functions, such as embryonic development, bone formation, reproduction, arthritis and cancer [5]. The LIV-1 family motif is unique in that it contains a novel conserved proline and an additional glutamic acid residue (indicated in bold), HEX**P**H**E**XGD (Figure 1 and Figure 2). In conventional metalloproteases, zinc co-ordinates with both histidine residues in the motif as well as a water molecule from the first glutamic acid, and other additional residues downstream, depending on the subgroup [13]. The LIV-1 family has no Met-turn of metzincins [14], but some conserved residues downstream may act as ligands. Using this knowledge, it has been previously proposed that the second glutamic acid or a histidine adjacent to TM3 domain may be involved in the zinc co-ordination [4], as shown in Figure 4. 

Zinc transporters ZIP8 and ZIP14 have the initial histidine of this motif replaced by a glutamic acid as well as the replacement of the conserved histidine in TM2 by a glutamine. These two residue changes make ZIP14 and ZIP8 ion transport less specific for zinc and provide the ability to transport many other ions. Both ZIP8 and ZIP14 have been recognized to transport cadmium, zinc, iron and manganese [15,16,17,18,19,20] and the transportation of ions other than zinc has also been observed in other ZIP family members, although the full physiological relevance of this is still undetermined. Mutation of individual residues within ZIP8 confirmed their involvement in the specific transport of cadmium and manganese, especially the glutamic acid residue that replaced the histidine in TM5 [21]. 

Although there is no available crystal structure for any human ZIP transporter, the crystal structure of a ZIP transporter from the bacteria *Bordetella bronchiseptica*, BbZIP, has been obtained [11]. This structure, which was obtained in the presence of cadmium ions, shows that TM2 and TM5 could be close together within the tertiary fold, consistent with a role for the conserved prolines in these transmembrane domains. The overall structure reveals an unusual 3 + 2 + 3 transmembrane arrangement. Additional modelling of this data without the presence of zinc confirms the likely involvement of TM2 and TM5 with the zinc binding site [22]. Furthermore, it is suggested that ZIP transporters may work as a two-domain elevator-type mechanism, previously observed in other channels [22] as well as providing a proposal for a dimer structure. The dimer would consist of a central region from each monomer consisting of TM2, TM3, TM7 and TM8 which would form the pore region while the zinc binding interaction would happen at either side of this pore and consist of TM1, TM4, TM5 and TM6. 

The substantial advancement in protein structure prediction for large and complex proteins including transmembrane proteins such the LIV-1 subfamily, achieved through the use of artificial intelligence and deep learning to develop AlphaFold [23] enables the predicted structures of these zinc transporters to be compared meaningfully. Looking at the AlphaFold predictions for ZIP6 and ZIP10 it is possible to see the eight TM domains, enclosed within the purple box, and the clearly defined CPALLY motif in the N-terminus of both ZIP6 and ZIP10 (Figure 5). ZIP7 and the other LIV-1 family member that does not reside on the plasma membrane, ZIP13, do not have this motif, consistent with the belief that it may be involved in interacting with the pore region and regulating travel across it at the plasma membrane. Additional roles of the CPALLY motif are discussed in Section 3.4 below. Intriguingly, both ZIP6 and ZIP10 have what looks like an extension to TM2 which protrudes at both sides of the TM domain. This extended helix is not present in other LIV-1 family proteins suggesting it may be relevant to the specific function of ZIP6 and ZIP10. As ZIP6 and ZIP10 are unusual amongst the LIV-1 family members in that they form a heteromer, it is interesting to suggest that this long similar region that only occurs in these two zinc transporters may be important for heteromer formation, perhaps providing a means for their interaction. 

### 3.2. Conserved Proline Residues within TM Domains

It is noteworthy that the proline residue in TM5 of the LIV-1 family of proteins is completely conserved, consistent with it having an important role in protein function. The significance of this conserved proline in the HEXPHEXGD motif, which can both stabilize and disrupt helices [24,25], has not been proven. More recently, proline residues within the transmembrane domains have been accepted [26], especially in proteins likely to act as channels which need a switch mechanism to help open and close the channel [27]. However, proline residues are well known to produce kinks in tertiary structures and this can be a useful property in an ion channel, providing a mechanism to help open and close the channel gate [27]. Since ZIP transporters move zinc and other ions down a concentration gradient they are likely to function more as channels that need to open and close the gate, providing an opportunity for the proline residue to be essential for the change in conformation required for ion flow. Phosphorylation is one mechanism that can aid in channel opening and this is mentioned in more detail in Section 4.2. 

Proline residues have also been implicated in helix interaction and protein stability within membranes [28] and are now known to play an important role in the structure and function of transmembrane proteins [29]. Interestingly, scanning protein mutations present in the Human Gene Mutation Database have revealed that proline mutations generate a high phenotypic propensity for disease [30], confirming their important role in protein function.

It is worth noting that the LIV-1 family of zinc transporters contain another conserved proline residue in TM2 which may also be involved in pore formation with TM5 in the membrane. Recent structural insights into the tertiary structure of the ZIP transporters suggests that TM2 and TM5 could interact to form the pore [11] and further analysis confirms this likelihood [22]. TM4 contains a conserved histidine not usually recognized in transmembrane domains and TM5 contains two conserved histidine residues as well as a proline residue which has been traditionally excluded from transmembrane domain prediction due to its ability to generate a kink in the tertiary structure. 

### 3.3. Histidine Residues

Histidine residues are relatively rare in globular proteins, occurring in only 2% of cases. However, between 30–50% of proteins are metalloproteins, requiring a metal ion for normal structure or function [31]. Histidine is the most common metal coordinating ligand [32] and therefore it is not surprising that proteins responsible for transporting zinc contain many histidine residues. The ZIP transporters contain histidine residues at three main positions in addition to those conserved within the transmembrane domains. The additional histidine residues are positioned in the N-terminus, the extracellular loop between TM2–3 and the intracellular loop between TM3–4. The LIV-1 family members contain many more histidine residues than the other ZIP transporters in these positions. Subfamily I and II proteins contain up to five histidine residues and ZIP9 from the Gufa family contains 12 histidine residues in the three loops, whereas the LIV-1 family members can contain many more histidine residues in these regions. For example, ZIP10 contains 85 histidine residues overall, ZIP6 contains 69 histidine residues and ZIP7 contains a total of 57 histidine residues overall. Some individual histidine residues have been mutated and linked to zinc transport function but currently the role of the all the histidine residues remains unexplored.

### 3.4. CPALLY Motif and Link to Prions

The LIV-1 family contains an unusual motif termed the CPALLY motif within their N-terminus [8]. This motif consists of 40 residues in total and fits the overall consensus sequence C (X26) CPALLYQ (X5) C, containing three conserved cysteine residues. X represents any amino acid allowed between the other residues. As cysteine residues form disulfide bonds with other cysteine residues, it is assumed that the third cysteine is available to bind with another cysteine residue in a different region of the protein sequence. Interestingly, only the LIV-1 family members that localize to the plasma membrane contain this CPALLY motif in their N-terminus and these members also contain a conserved cysteine residue immediately before the HEXPHE motif in TM5. Therefore, it is speculated that this CPALLY motif may be involved in controlling access to the pore by binding to this cysteine in TM5 to prevent access. Whether this binding can block zinc influx has not yet been proven.

Interestingly, prion proteins have been shown have descended from the LIV-1 family of zinc transporters, in particular ZIP6, ZIP10 and ZIP5 [33], which are all on the same arm of the family phylogenetic tree [10]. Prion proteins are known to bind zinc and then internalize it at concentrations well below physiological levels of zinc [34]. The N-terminus of ZIP6 and ZIP10 have a comparable tertiary structure to that of prion proteins [35] aided by similar essential disulfide bonds. The CPALLY motif in the LIV-1 family is positioned in the same region as the disulfide bonds in prion proteins [33], producing an almost identical tertiary structure for ZIP6, ZIP10 and prion proteins which is consistent with these proteins exhibiting a common mechanism.

Prions are also known to bind to NCAM1 and play a role in the mechanism of cell adhesion and detachment [36]. This is an important discovery, especially as both ZIP6 and ZIP10 are known to be able to initiate cell rounding and detachment, leading to epithelial and mesenchymal transition (EMT) [10]. Both ZIP6 and ZIP10 have been implicated in metastatic cancers [37,38,39] which is thought to be due to their ability to influx zinc, inhibit GSK3-beta and activate the transcription factor Snail to turn off the production of adherence genes, such as E-cadherin [40]. This mechanism ignores the additional fact that the extracellular N-terminus of ZIP6 and ZIP10 can bind to NCAM1, due to its similar shape to that of prions, allowing zinc influx to activate cell detachment [10]. 

After the zinc influx into the cell, the large number of histidine residues in the cytoplasmic region of the ZIP6/ZIP10 heteromer (Figure 1) would likely be able to bind zinc and keep it close to the juxtamembrane region of the heteromer, allowing it to specifically inhibit GSK3-beta activity in that region and provide a means for NCAM1 to dissociate from the extracellular milieu of proteins, thus resulting in cell detachment [10]. Whether this exact mechanism is used by cells to detach from tumors to metastasize has not yet been determined. 

## 4. The Role of Post-Translational Modification in LIV-1 Family Function

Over twenty years since the LIV-1 family was first documented [4] there is still a paucity of information regarding the role of the post-translational modification of these proteins and the role it plays in their function. However, both proteolytic cleavage and phosphorylation have been demonstrated to be functionally important for some transporters.

### 4.1. Proteolytic Cleavage

Some LIV-1 family members have been demonstrated to be proteolytically cleaved as part of their functional activity, although no proteases have been designated to this role. Cleavage of the ectodomain of ZIP4 has been demonstrated during zinc deficiency [41] and ectodomain shedding is thought to be an important regulatory mechanism for the ZIP4 protein. ZIP10 has also been shown to be cleaved at its N-terminus [42] as confirmed by a variety of band sizes when analyzed by Western blotting but the relevance of this cleavage to protein function is still uncertain. Two members of the LIV-1 family, ZIP6 and ZIP10, both contain a highly predicted PEST site in the middle of their N-terminus [43]. These sites are rich in P, E, S and T residues and predict a strong likelihood of proteolytic cleavage. Cleavage of ZIP6 in the N-terminus consistent with this cleavage site has been recognized as a requirement for the ZIP6 protein to relocate to the plasma membrane [40]. This was confirmed by the failure to recognize the far N-terminal region of ZIP6 at the plasma membrane despite it being present in the endoplasmic reticulum [40], presumably as the full-length protein. In addition, ZIP10 is also predicted to have a strong PEST cleavage site between TM3–4, suggesting the ability of the protein to be cleaved in the middle of the TM region. If this were the case, it may suggest a mechanism for quickly destroying the zinc transport’s ability and/or an ability to degrade fragments of the molecule once no longer useful. 

It is intriguing to consider some families of proteases called Site 2 proteases (S2P) which have the ability to cleave other proteins especially within their TM domains [44]. These proteins usually have an HEXXH motif in one direction within a TM domain and an additional motif, such as LDG, in a different TM domain travelling in the same direction (Table 1), as indicated by arrows. The motifs in the protease families in the bottom three rows are always in the same direction within the membrane. 

Additionally, the signal peptide peptidase (SPP) family of proteases also contain TM motifs although they do not have the HEXXH motif. Interestingly, these proteases also contain a motif in their C-terminal region which has similarities to the CPALLY motif in ZIP transporters. The ZIP transporters all contain motifs within different TM domains which appear to be organized in the same direction to each other (Table 2) and contain similar amino acids to those seen in the protease motifs. Whether ZIP transporters can utilize these motifs to actually cause proteolytic cleavage of themselves or other proteins in a zinc-dependent manner has not been investigated. 

### 4.2. Phosphorylation

The LIV-1 family members reside in the plasma membrane or in intracellular membranes, bringing zinc into the cytoplasm from these locations [10]. The intracellular free zinc is believed to be in the broad region of 250 pM, yet the total cellular zinc is hundreds of micromolar [45]. This means that the concentration of zinc within cellular compartments, such as the endoplasmic reticulum, is considerably higher than in the cytosol and therefore, the transport of zinc across these channels either from stores or extracellularly into the cytosol would be down a zinc concentration gradient. Transport of ions through a channel and down a concentration gradient would require a mechanism for the channel gate to open, such as phosphorylation. 

In an effort to investigate whether ZIP transporters were in fact activated by phosphorylation we used ZIP7 as an example. ZIP7 is located on the endoplasmic reticulum and is responsible for releasing zinc out of that store [46]. Having used phosphorylation prediction software to predict four potential phosphorylation sites on the large intracellular loop between TM3–4, we observed that two of these sites, residues S275 and S276, were both predicted to be phosphorylated by protein kinase CK2 and also had multiple hits by mass spectrometry, suggesting that they were phosphorylated [47]. Activation of ZIP7, by CK2 phosphorylation of residues S275 and S276, to transport zinc from the endoplasmic reticulum was demonstrated using recombinant constructs of ZIP7 that had these key residues removed [47]. Furthermore, by using phosphokinase arrays to investigate the downstream activation effects of activated ZIP7, we established that ZIP7-mediated zinc release activated multiple signaling pathways known to lead to cell proliferation and growth, such as MAPK, AKT, PI3K, mTOR and the inhibition of GSK3-beta [46]. Additionally, using a monoclonal antibody that only recognized ZIP7 when phosphorylated on S275 and S276, increased phospho-ZIP7 was confirmed in aggressive breast tumors [48], suggesting the potential for this antibody to be used as a biomarker in cancer. 

These data indicate that ZIP7 acts as a channel to transport zinc from intracellular stores to the cytosol and the channel gate can be opened by phosphorylation to allow zinc to move down a concentration gradient. The cytoplasmic loop region between TM3–4 contains mixed charged residues and is not conserved between family members, suggesting the potential for differential regulation between transporters, especially as there are numerous predicted phosphorylation sites between TM3–4 for most of the other LIV-1 family members (Figure 6). 

Initial investigations with ZIP6 and ZIP10 have suggested that there are indeed multiple kinases involved in their activation which now need further study. It seems likely that other LIV-1 family members also require phosphorylation to open the channel gate and definition in this area will enhance future investigations. The kinase predictions for these sites are extremely diverse and consistent with each transporter having a different and independent role. Interestingly, apart from serine and threonine predictions there are also tyrosine predictions for ZIP6, ZIP10 and ZIP14. One predicted tyrosine site in ZIP6 has been observed 126 times in mass spectrometry studies, suggesting that tyrosine phosphorylation may also be important in channel function. However, to date there are no studies investigating the role of tyrosine phosphorylation on zinc transporter function.

Interestingly, histidine residues have also been demonstrated more recently to have a phosphorylation capability [51]. Histidine residues are usually rare in proteins; however, the LIV-1 family of zinc transporters have an abundance of histidine residues, with ZIP10 having 76 histidine residues in three extra membrane loop regions with additional histidine residues present in the transmembrane domains. Whether the histidine residues in the ZIP family of zinc transporters also play a role in their activation by phosphorylation has not been investigated.

## 5. The Role of ZIP Transporters in Cancer

It is well known that zinc is essential for normal cell growth and proliferation [45]. The zinc released from the ER by zinc transporter ZIP7 has been demonstrated to activate many intracellular signaling pathways leading to cell growth and proliferation [46], such as AKT [52], PI3Kinase [53], MAPK [54] and mTOR [55] and to inhibit GSK3-beta [56], also leading to growth. Therefore, this means that cellular zinc increases above the normal values are likely to increase cell growth and proliferation and need to be controlled to prevent aberrant levels that would encourage diseases of excess growth, such as cancer. 

Recently, several LIV-1 family zinc transporters have been connected to driving the growth of cancer cells. Some of these zinc transporters have more of a defined role than others and the following discussion is restricted to three LIV-1 family members that have been investigated in detail. 

### 5.1. ZIP7 Drives Anti-Hormone-Resistant Breast Cancer

Many estrogen receptor-positive breast cancers are often treated with anti-hormones. One commonly used anti-estrogen agent is tamoxifen. Unfortunately, although tamoxifen treatment can prevent or slow the growth of the cancers this is not always permanent and with time most cancers become resistant to tamoxifen by developing more aggressive phenotypes. Our wider group has developed a tamoxifen-resistant cell model from MCF-7 breast cancer cells that we have previously characterized for zinc signaling mechanisms [48]. During this investigation we observed that the tamoxifen-resistant cells had a large increase in ZIP7 as well as a substantial increase in the level of intracellular zinc. These effects could explain this new phenotype of tamoxifen resistance, especially since zinc is well known to inhibit tyrosine phosphatases [57], thereby allowing the tyrosine kinases used by these tamoxifen-resistant cells to activate alternative tyrosine kinase receptor pathways such as EGFR and IGF-1R which are activated for growth [58].

ZIP7 requires phosphorylation on residues S275 and S276 before it can release zinc from stores [59]. As a result an antibody has been generated that only recognizes the active phosphorylated form of ZIP7 [46] and this antibody can be used to recognize higher grade breast cancers [48] which have greater levels of ZIP7. This antibody may provide a useful assessment of cancers prior to treatment.

### 5.2. ZIP6 and ZIP10 Are Essential for Cell Mitosis

ZIP6 and ZIP10 transporters have always behaved differently to the other LIV-1 family transporters that we have investigated. Primarily this has been due to the fact that they both play an important role in initiating migration through cell rounding and detachment [40]. This then means that in laboratory culture, cells expressing ZIP6 or ZIP10 may have already detached from the dish before harvesting, making it difficult to examine their effects. In contrast, due to the recent discovery of the essential role of ZIP6 and ZIP10 in initiating mitosis [60], when mitosis is stimulated in cells, using agents such as nocodazole, both endogenous ZIP6 and ZIP10 are much easier to find. They both seem to be highly regulated by the cell, meaning that they are both degraded, probably by ubiquitination, during the G and S phases of the cell cycle so that they do not allow the cell to start mitosis prematurely. In order to initiate mitosis, the ZIP6/ZIP10 heteromer first has to move to the plasma membrane by cleavage of ZIP6 at the N-terminus [60]. Once positioned on the plasma membrane, the ZIP6/ZIP10 heteromer influxes zinc which binds to pS^705^STAT3, the transcription active form of STAT3, and changes it to pS^727^STAT3 which continues to bind to the ZIP6/ZIP10 heteromer throughout mitosis [60]. The large number of histidine residues in the cytoplasmic loops of ZIP6 and ZIP10 (Figure 1) will allow this to happen in the close vicinity of the heteromer and plasma membrane and generally prevent the spread of zinc further into the cell. The pS^727^STAT3 form also binds to pStathmin during mitosis [61], the form of Stathmin that allows microtubule rearrangement as required in mitosis [62]. At the end of mitosis the C-terminal end of STAT3 is cleaved, physically removing residue S727 and allowing the normal transcription active form of STAT3 to drive a new round of the cell cycle. 

This important discovery can be used to inhibit the growth of cancer cells and, in the laboratory, the growth of many different types of cancer cells have already been inhibited using a ZIP6 or ZIP10 antibody which binds to the extracellular N-terminus of the ZIP6/ZIP10 heteromer, thus preventing the influx of zinc required for mitosis [60]. This work is currently being expanded for clinical development as a new anti-mitotic agent for cancer therapy. 

### 5.3. Human Tumour Database Evidence of the Role of ZIP Transporters in Breast Cancer

There are many useful databases emerging now that allow gene searching in clinical cancer samples. We have used Gepia [63] to search for change in gene expression for all nine LIV-1 family members (Figure 7) comparing their levels in cancer samples (red) to those in normal tissues (grey). Gepia uses data from the Cancer Genome Atlas (TCGA) [64] and Genotype-Tissue Expression (GTEx) [65] projects where RNA-Seq data was produced for tens of thousands of cancer and non-cancer samples. TCGA alone has produced RNA-Seq data for 9736 tumor samples across 33 cancer types, in addition to data for 726 adjacent normal tissues. Only three of these ZIP transporters showed a statistically significant increase in cancer tissue, namely ZIP4, ZIP6 and ZIP7. Additionally, ZIP10 showed a trend of increased levels in cancer which was not significant. It is not surprising due to our current functional knowledge that ZIP6, ZIP7 and ZIP10 should be increased in cancer tissues as they all have a key role to play in driving cell growth and/or cell division. 

Measuring zinc levels in the blood of cancer patients has proven to be very unreliable but the general consensus suggests that cancer tissues, with the exception of prostate cancer, do have elevated zinc levels. ZIP4 is known to be increased when cells are zinc-deficient due to its role in zinc uptake from food, suggesting the presence of a zinc deficiency in breast cancer cells causes an elevation in ZIP4 levels. This is an interesting result as both lower zinc serum levels and increased ZIP4 tissue expression have already been linked to a poorer prognosis in colon cancer [66], suggesting a role for ZIP4 in other cancers as well.

## 6. Additional Areas of Investigation

Despite the information above there are some areas where zinc transporters may play an important role yet they have not been thoroughly investigated and merits consideration in the future. These are mentioned below.

### 6.1. ZIP7 in Immunity and B Cell Activation

ZIP7 is known to release zinc into cells from the ER and this cytoplasmic zinc can be used to drive multiple growth and proliferation pathways [46]. Zinc is required for many functions in cells, including B cell activation [67]. Interestingly, ZIP7 is situated on the MHC class II region of the genome, suggesting it may have an important role in immunity. Prior to a positive immune response in cells, many required cofactors are bound tightly together in an inactive form and anchored to the plasma membrane by binding the membrane receptor CD40 [68,69]. Mass spectrometry experiments have shown that ZIP7 and CD40 are bound together [70] even though CD40 is present on the plasma membrane and ZIP7 is on the endoplasmic reticulum. It is interesting to speculate that ZIP7 may be part of this inactive immune complex, ready to be re-activated when the correct stimulus appears and allows these molecules to dissociate and activate. Interestingly, TRAF2 and TRAF6 are two of the molecules in this inactive immune complex and both are predicted to bind ZIP7 according to the ELM website [71]. TRAF2 is predicted to bind ZIP7 across residues S275 and S276, the two residues required for activation by phosphorylation [59], suggesting a TRAF2 bound ZIP7 would be inactive. Furthermore, TRAF6 is predicted to bind ZIP7 on the HEXPHE motif in TM5, thought to be involved in zinc transportation and again suggesting an inactive form of ZIP7 when bound to TRAF6. These data together suggest that ZIP7 may play an important role in the immune response, when bound to TRAF2, TRAF6 and CD40 as part of the inactive complex and when these disassociate to activate the immune response they allow ZIP7 to actively increase the zinc concentration to encourage proliferation. Increased activation of ZIP7 and increased zinc have both been observed in activated B cells [67] confirming a role for ZIP7. It has also been recently discovered that some mutations in ZIP7 can cause a lack of B cells and a subsequent lack of immunity [72], adding weight to the involvement of ZIP7 in the immune process. 

### 6.2. The Effect of ZIP Knockdown on Other ZIP Transporters

The LIV-1 protein family contains nine human members and individual transporters are often genetically manipulated to provide functional information. However, because there are nine family members, this may provide an opportunity for individual family members to be altered in cells to compensate when others are deleted. For example, we have used a mouse breast cell line that has had ZIP6 removed by CRISPR technology [36]. We have observed in this cell line that the level of ZIP10 increased dramatically, apparently to compensate for the loss of ZIP6 [60]. This makes sense, especially if the ZIP6/ZIP10 heteromer is essential for cell division, as the removal of ZIP6 may prevent cell division and cause cell death. It is assumed that the cell can compensate for the loss of ZIP6 by utilizing a ZIP10 dimer instead of the ZIP6/ZIP10 heteromer to initiate cell division and allow survival, albeit at a slower rate [60]. It is currently unknown whether ZIP6 or ZIP10 are equal partners in the complex that initiates mitosis or whether one is more dominant than the other.

This is a good example of when using genetic manipulation to alter a zinc transporter protein, it is important to assess any effects that this may have had on the other family members, to rule out any compensation by other transporters, before drawing premature conclusions of the effects of removing that individual protein. 

### 6.3. Zinc and Calcium

Increasingly, calcium channels have been shown to be able to transport zinc into cells. There is good evidence that zinc can be transported into cells by acetylcholine channels, glutamatergic receptors, voltage-gated calcium channels [73] and also members of the TRP family of calcium channels, especially TRPM7 [74]. Many calcium effects in cells happen in milliseconds whereas zinc effects take seconds or minutes. When zinc was first shown to be a second messenger [75] it was demonstrated that the zinc release from stores was downstream of a calcium release. Additionally, ZIP7 could be stimulated by zinc to release zinc from stores but it could equally be activated by EGF and a calcium ionophore [59], simulating calcium release, confirming that a calcium release is required upstream of a ZIP7-mediated zinc release. 

Other robust data exist which demonstrates that zinc levels in the endoplasmic reticulum can stimulate the ryanodine receptor to release zinc [76,77], providing another important functional link between the two ions. Perhaps the zinc treatment that stimulates the activation of ZIP7 to release zinc from stores is able to cause a calcium release from stores by stimulating the ryanodine receptor in the time needed to activate ZIP7. The relationship between calcium and zinc has not been fully examined and may provide a fruitful area of investigation in the future.

## 7. Conclusions

Although the LIV-1 family of SLC39A zinc transporters was originally documented over 20 years ago, there is still much to discover about the functions of these individual transporters. Furthermore, as there are nine human members it is likely that each transporter has independent functions, each regulated by different agents, providing different endpoints. This is certainly true for what has been discovered for ZIP7, ZIP6 and ZIP10 to date. Continued efforts in this area are likely to produce important information, especially as aberrant levels of these zinc transporters are increasingly being implicated in different disease states.

## 8. Patents

Some of the work reported in this manuscript is covered by a patent entitled ‘anti-mitotic composition comprising antibodies against ZIP6 and/or ZIP10’ with an international publication number of WO 2020/249483.

## Figures and Tables

**Figure 1 ijms-24-01255-f001:**
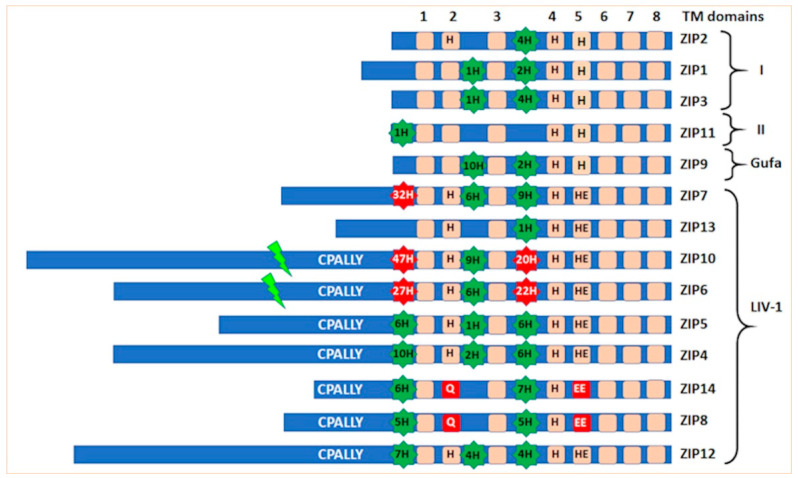
Comparison of features of the 14 human ZIP family members. The 14 human ZIP family members can be divided into four sub-groups with the LIV-1 family containing nine members. The highly conserved HEXPHEGD motif in TM5 of the LIV-1 subfamily is shown by HE. Conserved histidine residues in TM domains are shown by H. The different residues for ZIP8 and ZIP14 in TM2 and TM5 are shown in red boxes. Regions of histidine residues in the N-terminus, extracellular loop TM2–3 and intracellular loop TM3–4 are shown with green backgrounds with 10 residues or less and with red backgrounds when in excess of 10 residues. The predicted PEST site for N-terminal cleavage of ZIP6 and ZIP10 is indicated by a green shard. The CPALLY motif is present in the N-terminus of all LIV-1 family members that can reside on the plasma membrane.

**Figure 2 ijms-24-01255-f002:**
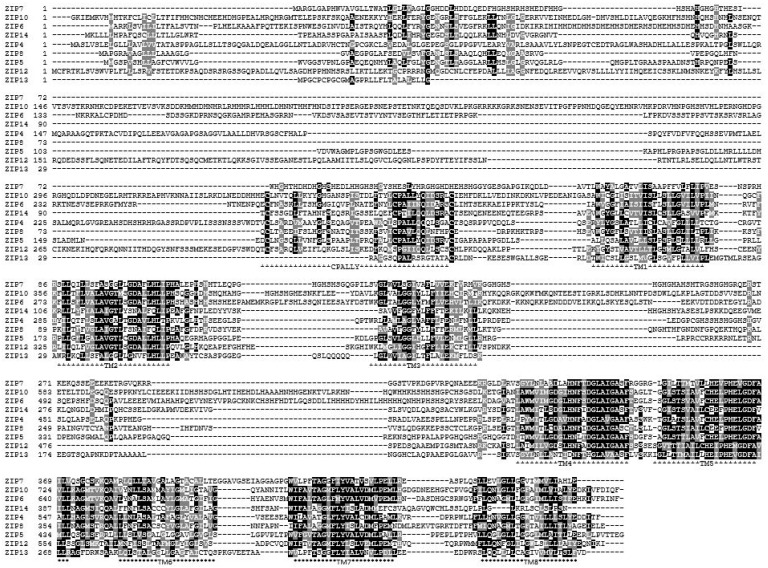
Alignment of the protein sequences of the LIV-1 family of ZIP transporters. All nine human members of the LIV-1 family of ZIP transporters have been aligned using Clustal Omega [12] and residues shaded using Boxshade. Residues in black represent identical residues and residues in grey represent similar amino acid families. The highly homologous eight transmembrane domains (TM) are identified by the high incidence of black shading indicating identical residues, especially within the HEXPHE motif in TM5. The CPALLY motif in the N-terminus is also indicated. The only large cytoplasmic loop, between TM3–4, has no homology, indicative of differential regulation for each protein.

**Figure 3 ijms-24-01255-f003:**
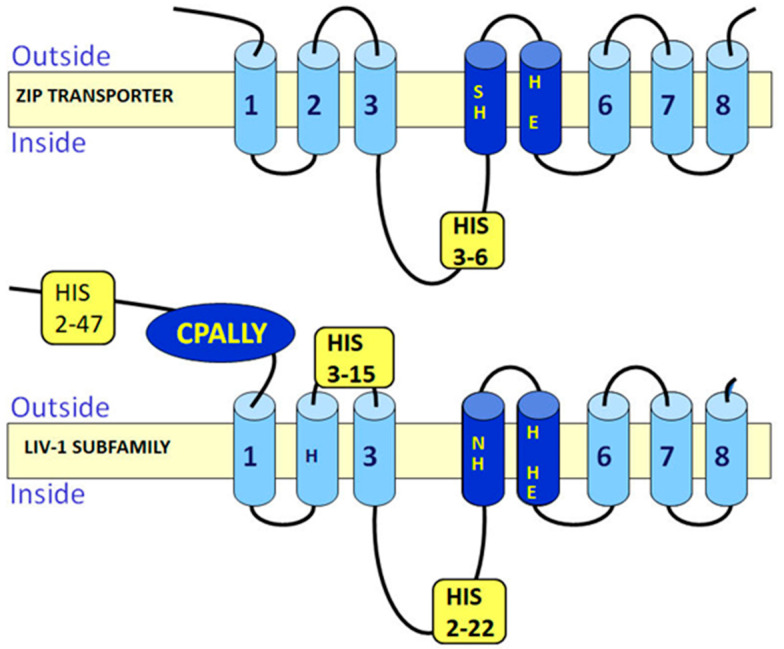
Schematic comparing the predicted structure of the LIV-1 subfamily of ZIP transporters with other ZIP transporters. The LIV-1 family contains the conserved HEXPHE motif in TM5 as well as a conserved His residues in TM2 and TM4. They also have many more His residues amassing 76 residues in ZIP10 alone which are spread between three regions, the N-terminus, the extracellular loop between TM2–3 and the intracellular loop between TM3–4. Additionally, the LIV-1 family members that move to the plasma membrane also have a conserved CPALLY motif on the N-terminus.

**Figure 4 ijms-24-01255-f004:**
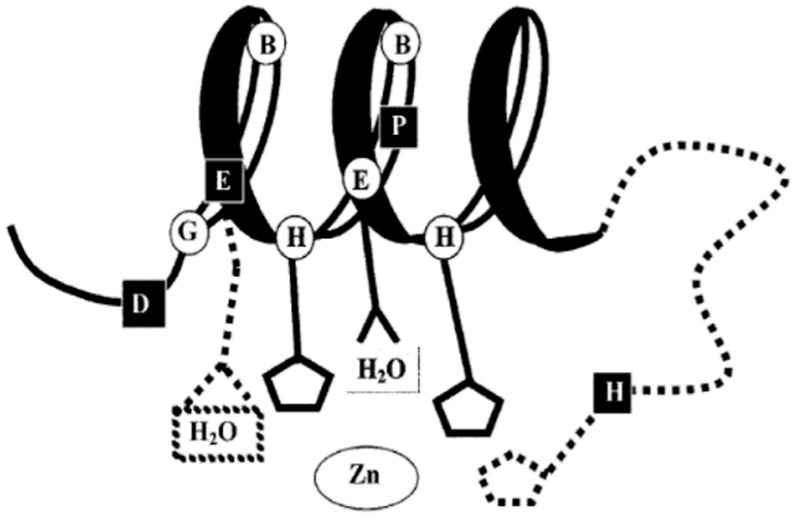
A schematic diagram of the zinc-binding environment of the metzincins has been altered to suggest the potential zinc-coordinating residues in the LIV-1 family. Residues shaded in black are known to coordinate zinc, square boxes represent conserved residues unique to the LIV-1 family, and dotted lines suggest the zinc-coordination ability. Figure taken from [4].

**Figure 5 ijms-24-01255-f005:**
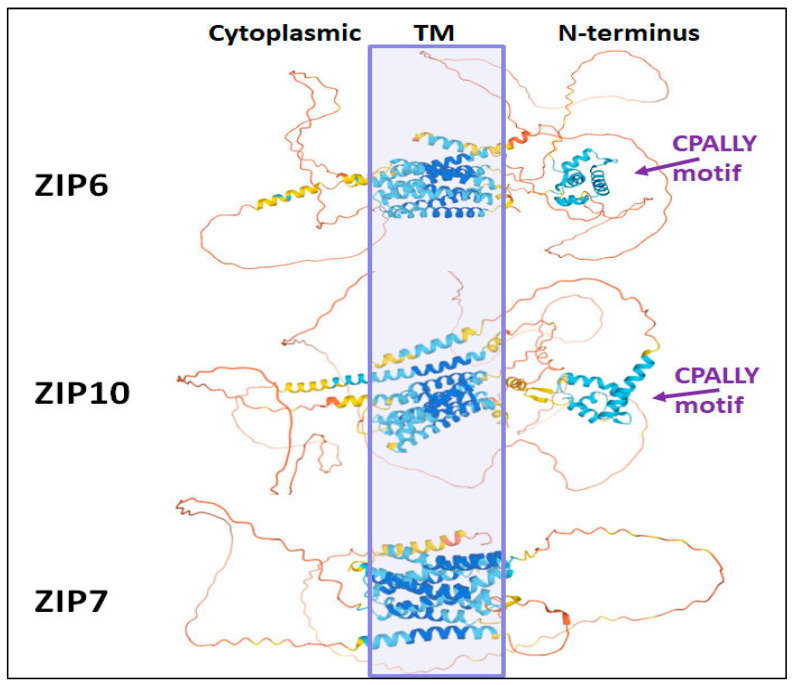
AlphaFold diagrams of three LIV-1 family members. AlphaFold diagrams of the predicted tertiary structures of three LIV-1 family members. ZIP6 and ZIP10 structures demonstrate the long extended TM2 region whereas ZIP7, which resides intracellularly, does not have this. The presence of the N-terminal CPALLY motif is also visible. The blue box roughly shows the position of the TM domains.

**Figure 6 ijms-24-01255-f006:**
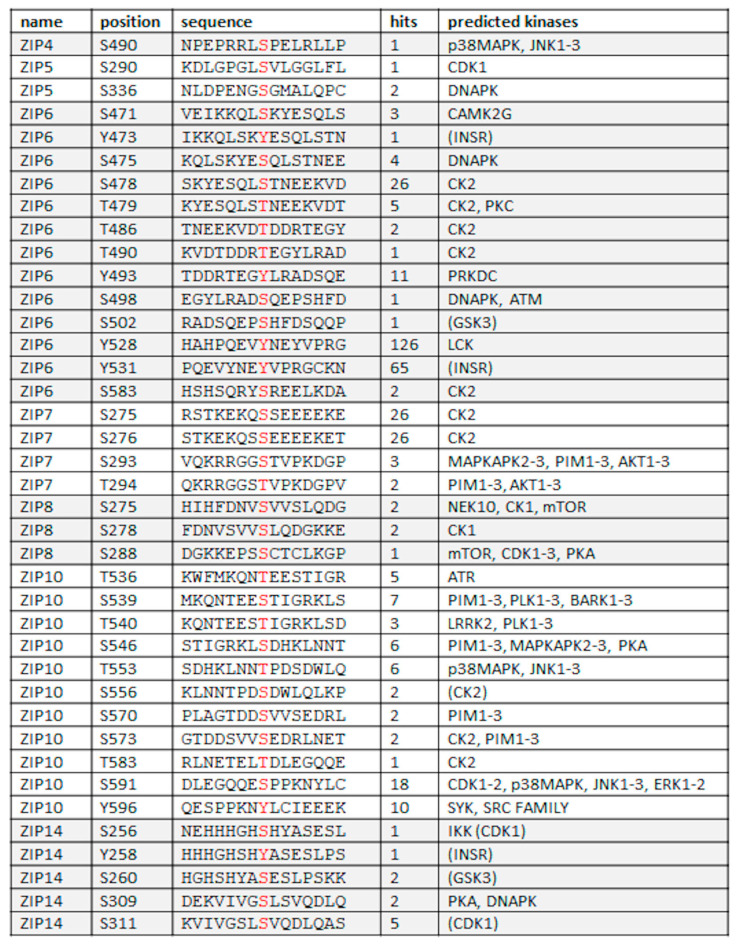
Predicted phosphorylation sites in the LIV-1 family of ZIP transporters. This table only includes the predicted sites on the cytoplasmic loop between TM3–4 and only those sites that were confirmed positive by mass spectrometry studies as detailed in the Phosphositeplus software [49] with occurrences listed in the hits column. Phosphorylated residues are shown in red. ZIP12 and ZIP13 are omitted as they did not have any predicted sites that met these criteria. Kinase predictions were taken from a number of prediction sites and those not reaching the cutoff in NetPhos [50] are shown in brackets.t.

**Figure 7 ijms-24-01255-f007:**
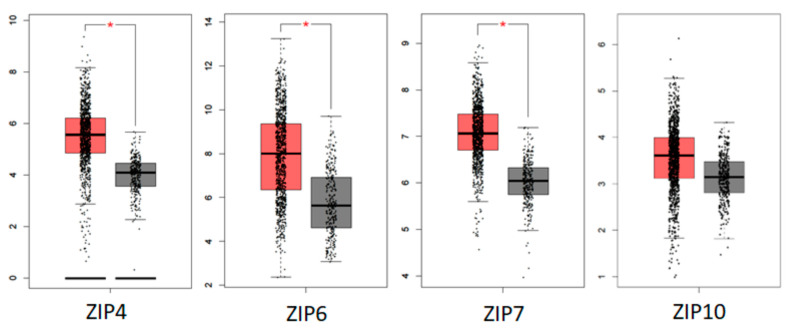
LIV-1 family members that are increased in invasive breast carcinoma tissues. This analysis was performed using Gepia [63] comparing the gene levels in 1085 tumors to 291 normal samples as transcripts per million. ZIP4, ZIP6 and ZIP7 all showed a statistically significant increase in breast tumor samples (red) compared to normal tissues (grey). ZIP10 levels also appeared to increase in the tumors but not significantly. The remainder of the LIV-1 family members did not show any increase and were excluded. * *p* < 0.05.

**Table 1 ijms-24-01255-t001:** Conserved TM motifs in protease families. The table below shows the conserved motifs in TM domains of five families of proteases as well as a C-terminal motif similar to the CPALLY motif in the N-terminus of the LIV-1 family of zinc transporters. The arrows indicate the direction of the motif within the TM domain showing consistency in each species.

Protease Family	TM Motif	TM Motif	TM Motif	TM Motif	C-Terminal Motif
Presenilin		YD 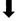	LGLGDFI 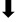		LPALPI
SPP		YD 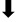	LGLGD 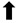		QPALLYXXP
S2P animals	HEIGH 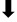		LDG 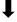		
S2P bacteria	HEIGH 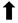		LDG 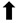		
spoIVFB	HEXXH 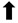		LDGG 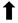	NLLP 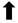	

**Table 2 ijms-24-01255-t002:** Conserved TM motifs in the LIV-1 family of zinc transporters. The table below shows the conserved motifs in TM domains of the nine human members of the LIV-1 family of zinc transporters as well as the N-terminal CPALLY motif. The arrows indicate the direction of the motif within the TM domain showing consistency in each protein, the down arrow represents the positioning from the outside towards the cytoplasm, whereas the up arrow represents the reverse direction.

Location + Direction	TM2 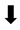	TM4 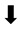	TM5 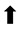	TM5 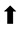	TM7 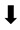	TM8 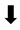	N-TERMINUS
ZIP4	HLTP	DGLA	HEXPH	LGDFA	HLTP	GLLXG	SPALLQQ
ZIP5	HLLP	DGLA	HEXPH	LGDFA	HLLP	GLLXG	CPALLYQ
ZIP6	HLLP	DGLA	HEXPH	LGDFA	HLLP	GMLXG	CPAIINQ
ZIP7	HLIP	DGLA	HEXPH	VDGFA	HLIP	GLLXG	
ZIP8	QLIP	DGLA	EEXPH	LDGFA	QLIP	GMLXG	CPAVLQQ
ZIP10	HLLP	DGLA	HEXPH	LDGFA	HLLP	GLLXG	CPALLYQ
ZIP12	HLIP	DGLA	HEXPH	MGDFA	HLIP	GLIXG	SPGIIQQ
ZIP13	HLLP	HGLA	HEXPH	VGDFA	HLLP	LLCXG	
ZIP14	QLIP	DGLA	EEXPH	LDGFA	QLIP	GLLXG	CPTILQQ

## Data Availability

Not applicable.

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
