# Peer review of "The LIV-1 Subfamily of Zinc Transporters: From Origins to Present Day Discoveries"

_ijms, 2023, doi:10.3390/ijms24021255_

Round 1

Reviewer 1 Report

This is a very clear and comprehensive review article that takes the reader from the origins of studies on this distinctive subfamily of zinc transporters through to cutting-edge work accomplished by the author, which includes well-cited work and very new, emerging findings. My comments are very minor.

The resolution of Fig 2 is very poor and would need to be sharpened before publication, I suggest. 

Zn2+ rather than Zn2+ (superscript) is used and should be corrected.

The turn of phrase used to introduce the reader to the section that compares the predicted structures of members of the family is rather out of line with the rest of the language, which assumes a ‘scientific’ rather than lay level of knowledge and familiarity.

Original text (L169) “Recently, artificial intelligence has been programmed to predict tertiary structures for proteins using currently available data. This is called alphafold and available online accessed from the Uniprot protein pages.”

Also, I would suggest that the applicability of AphaFold to prediction of the structure of TM proteins is validated.

Suggested alternative wording (with suitable citation) is:

The substantial advancement in protein structure prediction for large and complex proteins including transmembrane proteins such the LIV1 family, achieved through the use of artificial intelligence and deep learning to develop AlphaFold (REF: Cell Mol Sci 2022 Jan 15;79(1):73.  doi: 10.1007/s00018-021-04112-1.)enables the predicted strictures of these zinc transporters to be compared meaningfully.  

Author Response

My response to both reviewers is in the document uploaded

Reviewer 2 Report

This review by Kathryn Taylor effectively describes the history of identification and study of the LIV-1 sub-family of transition metal transporters. At the same time, there is quite a bit speculation throughout the paper, which lowers enthusiasm as it dilutes from currently known data. Overall, the review is well written, however as described below some adjustments could be made to improve the rigor of the text.

Concerns

The title of this review and at various parts throughout the text implies that LIV-1 transporters solely transport zinc. However, this is not the case. LIV-1 proteins can transport a wide variety of transition metals, and this should be indicated.

Outside of the separation of the four-subfamilies of SLC39A proteins, it is not clear how the comparison on Figure 1 is put together. Has this been done by sequence homology? Perhaps this can be better explained/discussed.

This reviewer did not find Figure 2 to be very helpful as it is not possible to read the sequences very well. As part of the description of the TM domains, it is suggested that some are not identified using bioinformatics. At this point, the 8 TM domains is well accepted, especially based on structural data. This should be noted.

Figure 3 notes that these proteins are channels. To date, this reviewer does not believe that there is any scientific evidence supporting this claim. Instead, SLC39A proteins are transporters, and this figure (as well as at other points in the manuscript) should be adjusted accordingly. The authors do correlate the presence of a TM proline as evidence that ZIPs are channels at the bottom of page 5. However, any direct evidence that ZIPs are a channel should be reported. Current data does not support the presence of a water-filled channel, which would presumably be required if ZIPs were ion channels.

On page 4, the author should better describe the Zincin and PDF groups. These terms are likely confusing for non-experts.

Throughout the manuscript, Zn2+ should have the 2+ superscripted.

In Figure 4, the author describes a putative zinc binding domain. However, there has been extensive work by multiple groups describing molecular modeling and direct structural determination of ZIPs. Some of this work has described two zinc binding sites. Other work has described oligomeric states of ZIPs. Unfortunately, most of this recent work seems to be neglected in this review. A better description of these findings would improve understanding of the structural basis of LIV-1s.

Figure 5 describes AlphaFold models of ZIP6, ZIP7 and ZIP10. Without a doubt, AlphaFold is a great advance and can be used to decipher structural information. However, the figure as presented here is not very informative. At the same time, there is direct structural information on the extracellular domain as well as the cytosolic loop. Integrating this data into the review would provide a better understanding of the structural differences between LIV-1s.

On page 6, the author notes that proline mutations are correlated with disease states. Are there any proline mutations within the LIV-1 sub-family that are correlated with diseases?

In section 3.2, the authors correlate the presence of histidine residues to metal coordination. Multiple groups have made many mutations of these histidine residues and followed their functional effects. At least some of these studies should be noted here.

In Table 1, the author refences the same direction and the opposite direction. What does this mean? When these domains are directly ion flux, presumably, in opposite directions are these domains oriented the same way in the membrane or is their directionality differing, in other words does flux go along these domains?

In the first paragraph of Section 4.2, the author compares different levels of zinc across different membranes. If ion flux was facilitated by concentration gradients, then the total zinc concentration is not very relevant. Instead, it would be the total free concentration, which would be more relevant. How does the free concentration of zinc in the cytosol compare to the free concentration of zinc in intracellular organelles such as the ER? Equally, ion flux for ion channels is driven by the electrochemical gradient. Does the electrochemical gradient support this direction of ion flow?

The authors used GEPIA to search for gene expression changes in cancer samples. It would be helpful to readers if the author could provide a more detailed experimental explanation. If the data is taken directly from reference 62, could the authors state this more explicitly.

In the conclusion, it is stated that as there are 9 human LIV transporters, then it is likely that that these proteins can have independent function. While this certainly could be true, it is probably important to mention that there are other members of this family that could also have independent or overlapping functions.

Author Response

My response to both reviewers is included in the uploaded dcoument.
